# Thiosemicarbazones Can Act Synergistically with Anthracyclines to Downregulate CHEK1 Expression and Induce DNA Damage in Cell Lines Derived from Pediatric Solid Tumors

**DOI:** 10.3390/ijms23158549

**Published:** 2022-08-01

**Authors:** Silvia Paukovcekova, Maria Krchniakova, Petr Chlapek, Jakub Neradil, Jan Skoda, Renata Veselska

**Affiliations:** 1Department of Experimental Biology, Faculty of Science, Masaryk University, 61137 Brno, Czech Republic; silvia.paukovcekova@mail.muni.cz (S.P.); maria.krchniakova@mail.muni.cz (M.K.); chlapek@sci.muni.cz (P.C.); jneradil@sci.muni.cz (J.N.); 2International Clinical Research Center, St. Anne’s University Hospital, 65691 Brno, Czech Republic

**Keywords:** thiosemicarbazones, anthracyclines, anthracenedione, pediatric solid tumors, combined anticancer treatment, checkpoint kinase 1, double strand breaks in DNA

## Abstract

Anticancer therapy by anthracyclines often leads to the development of multidrug resistance (MDR), with subsequent treatment failure. Thiosemicarbazones have been previously suggested as suitable anthracycline partners due to their ability to overcome drug resistance through dual Pgp-dependent cytotoxicity-inducing effects. Here, we focused on combining anthracyclines (doxorubicin, daunorubicin, and mitoxantrone) and two thiosemicarbazones (DpC and Dp44mT) for treating cell types derived from the most frequent pediatric solid tumors. Our results showed synergistic effects for all combinations of treatments in all tested cell types. Nevertheless, further experiments revealed that this synergism was independent of Pgp expression but rather resulted from impaired DNA repair control leading to cell death via mitotic catastrophe. The downregulation of checkpoint kinase 1 (CHEK1) expression by thiosemicarbazones and the ability of both types of agents to induce double-strand breaks in DNA may explain the Pgp-independent synergism between anthracyclines and thiosemicarbazones. Moreover, the concomitant application of these agents was found to be the most efficient approach, achieving the strongest synergistic effect with lower concentrations of these drugs. Overall, our study identified a new mechanism that offers an avenue for combining thiosemicarbazones with anthracyclines to treat tumors regardless the Pgp status.

## 1. Introduction

Anthracyclines belong to a class of natural cytotoxic antibiotics and represent highly potent antineoplastic agents widely used to treat a variety of tumor types [1]. Since the early 1970s, when the first compounds from the anthracycline class—doxorubicin (DOX) and daunorubicin (DAU)—were approved for clinical use, anthracyclines have been therapeutically used as potent anticancer drugs [1,2,3]. Their cytotoxic and cytostatic effects result from the combination of different mechanisms, i.e., intercalation into DNA, induction of oxidative stress, and poisoning of the topoisomerase II enzyme (TOP2) [4,5]. In general, DOX is usually indicated for treating lymphomas, sarcomas, and other solid tumors (bone tumors, lung carcinoma, bladder carcinoma, breast carcinoma, and cervical carcinoma), whereas DAU is more active on lymphoblastic and acute myeloblastic leukemia [1].

Despite its efficacy, anthracycline therapy is associated with cumulative dose-dependent cardiotoxic side effects, such as ventricular contractility and cardiomyopathy, leading to heart failure [6]. The other common side effects include acute nausea and vomiting, stomatitis, gastrointestinal disturbances, baldness, alopecia, neurologic disturbances (dizziness, hallucinations, and vertigo), and bone marrow aplasia [7]. Because of this toxicity, anthracyclines are often used in lower concentrations as a component in combination therapies [3]. To improve the therapeutic profile of anthracyclines, a synthetic anthracenedione—mitoxantrone (MIT)—was developed. This drug is routinely used in the treatment of lymphomas, leukemias, breast carcinoma, and prostate carcinoma [8].

In addition to the cardiotoxicity of anthracyclines, chemoresistance is a major reason for the failure of anthracycline therapy [9]. In general, the emergence of resistance to anthracyclines in malignant cells is a multifactorial process [9]. One of the main mechanisms is often associated with the acquisition of multidrug resistance (MDR) conferred by overexpression of permeability glycoprotein-1 (Pgp) [10,11,12]. Pgp is an efflux pump induced by drugs, heat shock, some natural products, environmental stress, and other nonspecific stress factors [9]. To date, several approaches have been explored to overcome MDR induced by Pgp [9], e.g., formulation of nanodelivery systems for anthracyclines [13,14,15], anthracycline derivatives with improved properties [16,17,18,19,20,21], and gene-targeted downregulation or pharmacological inhibition of Pgp [22,23,24,25,26].

Another promising strategy for overcoming Pgp-mediated drug resistance is based on combining DOX with thiosemicarbazones—di-2-pyridylketone 4-cyclohexyl-4-methyl-3-thiosemicarbazone (DpC) and di-2-pyridylketone 4,4-dimethyl-3-thiosemicarbazone (Dp44mT)—to directly leverage lysosomal Pgp transport activity [27,28]. Previously, Pgp was shown to be topologically inverted during the endocytic process, thus facing into lysosomes, leading to lysosomal loading of Pgp substrates [29,30,31]. Therefore, in cells with high Pgp expression, Pgp mediates not only drug efflux through the plasma membrane [32], but also increased drug trapping within lysosomes, creating drug “safe houses”, which can be observed [29]. DOX, DAU, and MIT were defined as lysosomotropic drugs on the basis of their pKa values, i.e., at acidic pH, these compounds are protonated and trapped in lysosomes, preventing their distribution to major targets in the nucleus [29]. DpC and Dp44mT are also Pgp substrates, and, in cells with MDR, they are transported via Pgp into lysosomes, where they are trapped due to protonation in a manner identical to anthracycline trapping [30,33,34,35]. However, positively charged thiosemicarbazones can bind copper and enter redox cycles to generate reactive oxygen species (ROS) that induce lysosomal membrane permeabilization (LMP) and subsequent proapoptotic signaling [27,30,33,34,35]. Moreover, LMP enables the release of stored anthracyclines from lysosomes, leading to restoration of their cytotoxic activity [27]. According to this mechanism, synergistic interactions between anthracyclines and thiosemicarbazones are expected to be beneficial predominantly in MDR malignant cells with typically high Pgp expression.

In this study, we focused for the first time on the combination of anthracyclines with thiosemicarbazones in pediatric solid tumors. Due to the improvement of therapeutic strategies, from aggressive surgical approaches to multimodal approaches with neoadjuvant chemotherapeutic treatments, almost 83% of children with a diagnosis of cancer become long-term survivors [36,37]. However, the use of new drugs has led to malignant acquisition of MDR, which is currently the leading cause of treatment failure in pediatric oncology [37]. Here, we aimed to address this problem and we present results that revealed a previously unrecognized mode of synergistic interaction between anthracyclines and thiosemicarbazones in model cell lines derived from pediatric solid tumors.

## 2. Results

### 2.1. SH-SY5Y Cells: The Only Tested Cell Line with High Pgp Expression

The first step in our study was to determine Pgp expression in untreated cell lines. For correct evaluation, two cervical carcinoma cell lines were used: KB-V1 cells with typical high Pgp expression were used as positive controls and KB-3-1 cells with minimal Pgp expression were used as negative controls. Analysis by qPCR revealed high *ABCB1* (the gene encoding Pgp) expression only in the SH-SY5Y neuroblastoma cells (Figure 1A). ABCB1 mRNA level in the SK-N-BE(2) neuroblastoma, Saos-2 osteosarcoma, and Daoy medulloblastoma cells was lower than it was in the SH-SY5Y cells, whereas ABCB1 expression in the RD rhabdomyosarcoma cell line was comparable with that in the negative control cells (Figure 1A). The results obtained using immunoblotting proved that the Pgp level was high only in the SH-SY5Y cells (Figure 1B, Appendix A). Moreover, to verify the obtained results, Pgp immunofluorescence staining was performed, and the results confirmed that Pgp signals were high only in the positive control and SH-SY5Y cells (Figure 2).

### 2.2. Thiosemicarbazones Acted Mainly Synergistically with Anthracyclines

The investigation continued with the quantification of the synergy between the thiosemicarbazones (DpC and Dp44mT) and anthracyclines (DOX, DAU, and MIT) using MTT assays with a subsequent analysis performed using Calcusyn software (Figure 3). The hypothesis on the synergy between these compounds was based on previously published findings [27,28] that described the synergistic effect of thiosemicarbazones and anthracyclines in overcoming Pgp-mediated multidrug resistance. 

First, the treatment was applied in the consecutive design treatment (Figure 3A). IC_50_ dose of DpC and Dp44mT (for 24 h) was determined according to the micromolar concentration (Table 1), whereas the IC_50_ dose of all the anthracyclines (used for 72 h) varied according to the nanomolar concentrations (Table 1).

The computational analyses of synergy showed that the combination of a thiosemicarbazone and anthracycline acted synergistically in four cell lines: SH-SY5Y, SK-N-BE(2), Daoy, and RD cells (Table 2). In the Saos-2 cell line, the analysis revealed moderate antagonism for each compound combination (Table 2).

In the next step, 200 nM Valspodar (VAL), a selective inhibitor of Pgp, was added to the consecutive design treatment to verify whether the mechanism of synergy was based on lysosomal sequestration of the compounds via Pgp (Figure 3B). We expected to observe a change in the compound interactions because of a synergistic, an additive, or even an antagonistic effect. The results did not confirm this expectation—synergistic effects were observed even after Pgp inhibition (Table 2). Moreover, treatment with VAL changed the moderate antagonistic effect of all compound groups in Saos-2 cells to a synergistic effect (Table 2).

As an additional analysis of anthracycline sequestration into lysosomes, fluorescence microscopy was performed. DOX is an autofluorescent compound, which allowed us to observe its effect on cells 2 h after its addition to the culture medium. For this purpose, the DOX concentration was increased to 10 µM. The lysosomes were visualized by indirect staining using an anti-LAMP-2 antibody. KB-V1 cells served as positive controls for the expected lysosomal sequestration of DOX (Figure 4A, Appendix A), whereas KB-3-1 cells served as negative controls (Figure 4B, Appendix A).

In general, this analysis showed that DOX did not colocalize with LAMP-2 but was primarily localized to the nucleus in all five tested cell types (Figure 4C–G, Appendix A–G). Therefore, we rejected the hypothesis that lysosomal sequestration of anthracyclines via Pgp played a major role in the synergistic effect of thiosemicarbazones and anthracyclines in the tested cell types derived from pediatric solid tumors.

Because the initial hypothesis explaining the mechanism of synergy was rejected, we changed the design to the concomitant treatment (Figure 3C), which allowed us to use markedly lower concentrations of thiosemicarbazones necessary to achieve the same effects as those achieved with micromolar concentrations in the consecutive design. The IC50 dose (for 72 h) for both DpC and Dp44mT varied in the nanomolar range (Table 1). Moreover, synergistic interactions between thiosemicarbazones and anthracyclines after concomitant treatment were observed in all five tested cell types (Table 2). Overall, the obtained data suggested that the concomitant design of treatment was more effective, enabling usage of lower drug doses than the consecutive design of treatment. Given the serious side effects associated with anthracycline therapy, approaches that would minimize concentrations of anthracyclines while maintaining their anticancer activity are needed. Therefore, in the subsequent analyses, we focused solely on the effects of the concomitant drug application given its apparent potential to reduce nonselective toxicity of the drugs.

### 2.3. Thiosemicarbazones Downregulated CHEK1 Expression, Leading to Mitotic Catastrophe after These Drugs Were Combined with Anthracyclines

As synergy between both thiosemicarbazones (DpC and Dp44mT) and all three anthracyclines (DOX, DAU, and MIT) was observed in all evaluated cell lines (SH-SY5Y, SK-N-BE(2), Saos-2, Daoy, and RD cells) after both combined treatments (consecutive and concomitant designs), the possible mechanism of synergistic interaction between these compounds seemed to be universal. Therefore, we predominantly focused on the common mechanisms of action of both types of compounds used.

The first of the main effects of anthracyclines is the production of double-strand breaks (DSBs) in DNA molecules, which are associated with cell cycle arrest in the G_2_/M phase. To confirm this effect in our tested cell lines, propidium iodide staining with subsequent flow cytometry analysis was performed to identify the changes in the cell cycle after 3 days of treatment with IC_50_ doses of DOX, DAU, and MIT. Cell cycle arrest in the G_2_/M phase was detected in four of the tested cell lines: SK-N-BE(2), Saos-2, Daoy, and RD cells (Figure 5, Appendix A). For the SH-SY5Y cells, only an increased proportion of cells in the sub-G_1_ phase, which is associated with formation of apoptotic bodies, was detected (Figure 5, Appendix A). Arrest in the G_2_/M phase would likely have been observed in an earlier phase of treatment, i.e., after 24 or 48 h.

For apoptosis induction based on the production of DSBs in DNA, it is essential to force cells with damaged DNA to bypass cell cycle arrest in the S/G_2_ phase and enter mitosis, which can lead to mitotic catastrophe [38,39]. Therefore, we investigated the ability of thiosemicarbazones, DOX, or their combinations to downregulate the expression of checkpoint kinase 1 and 2 (CHEK1 and CHEK2) at IC_50_ doses after 3 days. Immunoblotting revealed that both thiosemicarbazones effectively decreased CHEK1 levels in the SH-SY5Y, Saos-2, and Daoy cell lines, and a slight decrease in CHEK1 levels was also observed in the SK-N-BE(2) and RD cells (Figure 6A–E, Appendix A–E). DOX reduced the levels of CHEK1 less effectively than the thiosemicarbazones in all of the tested cell lines, and combined treatment caused a greater decrease in CHEK1 levels compared with single-compound treatments only in the SH-SY5Y cells (Figure 6A, Appendix A). In contrast, CHEK2 levels were almost completely abrogated in all the tested cell lines after treatment with DOX (Figure 6A–E, Appendix A–E). DpC and Dp44mT also decreased the CHEK2 levels in SH-SY5Y, SK-N-BE(2), Saos-2, and Daoy cells but less effectively than DOX (Figure 6A–D, Appendix A–D). The combined treatment did not achieve any enhanced effects considering the very strong inhibitory effect of DOX (Figure 6A–E, Appendix A–E). Regarding the obtained results, we can assume that downregulation of both checkpoint kinases by combined treatment with thiosemicarbazones and DOX can lead to the effective abrogation of the cell cycle checkpoint and the subsequent induction of mitotic catastrophe in cancer cells with damaged DNA.

### 2.4. Thiosemicarbazones Can Induce DSBs in DNA

In addition, the ability of thiosemicarbazones to produce DSBs in DNA (similar to anthracyclines) was investigated as a possible mechanism contributing to combination treatment synergy. The generation of DSBs after the selected treatment was verified by indirect immunofluorescence staining of the phosphorylated γ-H2AX protein, which is considered a biomarker of DSBs [40]. Staining was performed after 2 days of treatment with DpC, Dp44mT, or DOX alone and with their combinations (DpC + DOX or Dp44mT + DOX). The results obtained after 2 days showed that the IC_50_ doses of thiosemicarbazones caused γ-H2AX activation in all tested cell lines (Figure 7A–E, Appendix A–E). In general, the rate of DNA damage was apparently more intense after DOX treatment than after thiosemicarbazone treatment (Figure 7A–E, Appendix A–E). Enhanced DNA damage after administration of the DpC + DOX and Dp44mT + DOX combinations, compared to that after each compound was administered alone, was observed only in the Saos-2 cells (Figure 7C, Appendix A). Nevertheless, this method demonstrated that even nanomolar concentrations of DpC and Dp44mT induce DSBs in DNA.

## 3. Discussion

The obtained results proved the synergism of the combined thiosemicarbazone and anthracycline treatments in all chosen cell lines derived from pediatric solid tumors but also demonstrated that overcoming drug resistance mediated by Pgp is not the main mechanism of the observed synergism, as initially hypothesized. Accordingly, another synergistic mechanism based on enhanced DNA damage and the inhibition of checkpoint kinases was suggested.

Considering previous studies that demonstrated a positive correlation between Pgp expression and the synergistic action of thiosemicarbazones in combination with DOX [27,30,33,41], we started the investigation with Pgp screening in the chosen cell lines derived from childhood tumors. High Pgp expression at the mRNA and protein levels was found only in the SH-SY5Y cells. In the other cell lines, Pgp expression was very similar to that in the negative control cells. Therefore, the synergy of anthracycline and thiosemicarbazone treatments was expected only in SH-SY5Y cells.

To evaluate interactions between compound combinations, we first used a consecutive design of treatment, which was reported as effective in the literature [27]. Contrary to the predictions, synergistic action of thiosemicarbazones and anthracyclines in combination was found for four of the five cell lines. Therefore, to verify the main role of Pgp in the synergistic mechanism, VAL, a selective inhibitor of Pgp, was added to the consecutive design of treatment. In this regimen for compound application, we expected to see a change from synergistic action to an additive effect or even antagonism. Interestingly, the synergy between the thiosemicarbazones and anthracyclines was found in all five cell lines. Moreover, the analysis using fluorescence microscopy demonstrating primary localization of DOX to the nucleus in all cell lines contributed to the definitive rejection of the initial hypothesis stating that the main synergistic mechanism in chosen cell lines is based on lysosomal sequestration of Pgp.

To explore the least toxic regimen of treatment, we decided to change the combined treatment design from a consecutive to concomitant application of the drugs. With this change, we were not only able to reduce the thiosemicarbazone IC_50_ doses from the micromolar to nanomolar level, but were also able to observe synergistic action of the compound treatments in all five cell lines. Therefore, it can be concluded that concomitant application is the most convenient administration regimen for the combined treatment of anthracyclines and thiosemicarbazones in the tested cell lines.

Since the suitable application of the selected compounds was determined, the next objective was to describe the possible mechanism of the observed synergy. The mechanism seemed to be universal because all the combinations of DpC or Dp44mT with DOX, DAU, or MIT acted synergistically in four different cell types (neuroblastoma, medulloblastoma, osteosarcoma, and rhabdomyosarcoma cells). Therefore, we focused on the common features of thiosemicarbazone and anthracycline mechanisms of action.

The cytotoxic activity of anthracyclines is exerted through several mechanisms. In addition to their ability to intercalate between base pairs of DNA and induce ROS generation, anthracyclines can inhibit TOP2 activity by covalently binding to the DNA–TOP2 complex [5]. The potential lethality of stabilized DNA–TOP2 complexes is markedly increased during DNA replication, when the replication fork attempts to traverse this structure and convert transient single- or double-strand breaks into permanent double-stranded fractures [42]. Stalled DNA replication activates ataxia telangiectasia, Rad3-related (ATR) kinase, and, subsequently, CHEK1, which leads to cell cycle arrest mediated by cyclin A-Cdk2 inhibition [43,44].

In addition, ROS generated by anthracyclines cause DSBs in DNA and activate ATM-dependent pathways [45]. ATM kinase phosphorylates CHEK2, resulting in the inhibition of cyclin E-Cdk2 complexes and subsequent cell cycle arrest [46].

There is evidence of cross-talk and functional redundancy between ATR-CHEK1 and ATM-CHEK2 after anthracycline treatment. It was proven that DNA damage caused by DOX elicits both check kinase pathways [47].

The stabilization of the DNA–TOP2 complex, manifesting as DNA breaks, was observed after treatment with Dp44mT in a breast cancer cell type [48]. According to the observed expression of γH2A.X—a specific reporter of DSBs [49,50,51], our results indicate that both DpC and Dp44mT caused DNA damage in the neuroblastoma, medulloblastoma, osteosarcoma, and rhabdomyosarcoma cell types. This analysis revealed that both thiosemicarbazones can induce DSBs but with much less efficiency than DOX.

One promising strategy for enhancing the effect of anthracyclines is based on their combined application with drugs targeting cell cycle checkpoints (CHEK1 and CHEK2). In general, cell cycle arrest is a major mechanism of cellular resistance to drugs that cause DNA damage [52,53]. Therefore, TOP2 inhibitors can be combined with drugs inhibiting the checkpoint kinase pathways and administered to force cancer cells to bypass cell cycle arrest and enter mitosis with DNA damage, which leads to mitotic catastrophe [38,39]. For drug combinations with DOX, it was previously demonstrated that the double inhibition of CHEK1 and CHEK2 did not achieve better efficacy than inhibition of CHEK1 alone. Thus, only inhibition of CHEK1 abolishes DOX-induced cell cycle arrest followed by mitotic catastrophe [54]. Our investigations proved that both DpC and Dp44mT can downregulate CHEK1 in all tested cell lines derived from the most frequent pediatric solid tumors.

Regarding the obtained results, we can assume that the observed synergy between the selected thiosemicarbazones and anthracyclines is based on the following mechanism. Both types of compounds, thiosemicarbazones and anthracyclines, cause DNA damage by inducing replication fork stalling and ROS production. Therefore, their combined treatment leads to an increase in DSB induction. Moreover, thiosemicarbazones decrease CHEK1 expression and, thus, help cells with damaged bypass arrest, increasing the rate of cells undergoing mitotic catastrophe. Together, our results suggest a new mechanism of synergism between thiosemicarbazones and anthracyclines that is effective in tumor cells regardless of the Pgp expression.

## 4. Materials and Methods

### 4.1. Chemicals

The selected thiosemicarbazones Dp44mT (Cat. No. SML0186) and DpC (Cat. No. SML0483), the DOX (Cat. No. D2975000), DAU (Cat. No. D0125000), and MIT (Cat. No. M2305000), and Pgp inhibitor VAL (Cat. No. SML0572) were obtained from Sigma-Aldrich (St. Louis, MO, USA). All chemicals were prepared as stock solutions in DMSO (Cat. No. D8418, Sigma-Aldrich) and then diluted in cell culture medium to achieve a DMSO concentration < 0.5% (*v*/*v*), which has been shown to have no effect on cell proliferation relative to the control medium [55]. To prepare the stock solution, DpC, Dp44mT, MIT, and VAL were dissolved to reach a concentration of 100 mM; similarly, the DOX and DAU concentrations were each 10 mM.

### 4.2. Cell Culture

RD rhabdomyosarcoma cell line (Cat. No. 85111502), SH-SY5Y (Cat. No. 94030304), and the SK-N-BE(2) (Cat. No. 95011815) neuroblastoma cell lines were purchased from the European Collection of Authenticated Cell Cultures (Salisbury, UK). The Daoy medulloblastoma cell line (Cat. No. HTB-186) and Saos-2 osteosarcoma cell line (Cat. No. HTB-85) were purchased from the American Type Culture Collection (Manassas, VA, USA). The KB-3-1 and KB-V1 cervical carcinoma cell lines were gifts from Prof. Nóra Kucsma (Szakács Gergely’s Laboratory, Budapest, Hungary). The selected cell lines were cultured in various culture media, as shown in Table 3, and maintained under standard cell culture conditions: 37 °C in an atmosphere of 95% air and 5% CO_2_. All reagents for cell culture were purchased from Biosera (Nuaille, France).

### 4.3. Treatment Protocol

MTT proliferation assays were used to determine the IC_50_ values for the anthracyclines and thiosemicarbazones in the SH-SY5Y, SK-N-BE(2), Saos-2, Daoy, and RD cell lines. The cells were seeded in 96-well plates at a density of 5 × 10^3^ cells/well (SH-SY5Y, SK-N-BE(2), Saos-2, and RD cells) or 2 × 10^3^ cells/well (Daoy cells) in 100 µL of complete DMEM, and then allowed to adhere overnight. Different seeding densities for the cell lines were chosen to ensure that the cells remained in the log phase of growth during all 3 days of treatment. The IC_50_ values for the individual compounds were calculated from the growth inhibition curves obtained after treatment with increasing compound concentrations based on the initial estimated IC_50_ values (⅛-, ¼-, ½-, 1-, 2-, 4-, and 8-fold of the estimated initial IC_50_). The IC_50_ values for both anthracyclines and thiosemicarbazones were obtained after 3 days of treatment. For thiosemicarbazones only, the IC_50_ values were also obtained after 1 day of treatment.

To determine the combination index (CI) value, three different experimental designs of compound application were chosen (Figure 3). The “consecutive design” (Figure 3A) included the pretreatment of cells with an anthracycline (DOX, DAU, or MIT) only. After 48 h, DpC or Dp44mT was added to the cells undergoing anthracycline treatment. Proliferation was evaluated after 24 h using MTT assay. The second experimental design, labeled “+ Valspodar” (Figure 3B), was the same as the “consecutive design” with the addition of 200 nM VAL along with the anthracycline in the first treatment step. In the “concomitant design” (Figure 3C), both anthracyclines and thiosemicarbazones were added in combination to the cells. Proliferation was evaluated using MTT assay after 72 h of culture. Growth inhibition curves for compound combinations were constructed for increasing drug concentrations based on previously determined IC_50_ values (⅛-, ¼-, ½-, 1-, 2-, 4-, and 8-fold of the previously determined IC_50_).

For direct observation of DOX administration in cells, the cells were seeded onto coverslips in Petri dishes (35 mm in diameter) and allowed to grow to 80% confluence at 37 °C. All the cell lines were then treated with 10 µM DOX and observed after 2 h of incubation at 37 °C.

An indirect immunofluorescence assay was employed to reveal DSBs in DNA by staining for phosphorylated histone γH2AX. The cells were seeded onto coverslips and allowed to grow to 80% confluence as described above. The cell lines were then treated with the respective IC_50_ doses of DOX, DpC, Dp44mT, or with their combinations (DOX + DpC or DOX + Dp44mT) and incubated at 37 °C for 2 days.

To analyze the cell cycle using propidium iodide staining, all the tested cell lines were seeded in Petri dishes (90 mm in diameter) at a density of 1 × 10^5^/dish. The cells were then treated with the respective IC_50_ doses of DOX, DAU, and MIT and incubated at 37 °C for 3 days.

Immunoblotting was performed to determine the levels of CHEK1 and CHEK2. The cells were seeded in the same manner as in the cell cycle analysis described above. The cells were then treated with the respective IC_50_ doses of DOX, DpC, Dp44mT, or with their combinations DOX + DpC or DOX + Dp44mT and incubated at 37 °C for 3 days. 

### 4.4. Cell Proliferation

A colorimetric MTT assay was performed to evaluate the cell proliferation rate as previously described [56]. Briefly, the treated cells were incubated with MTT (0.5 mg/mL; Sigma-Aldrich) at 37 °C for 3 h. The formed formazan crystals were dissolved in 200 μL of DMSO. The absorbance was read at 570 nm with a reference absorbance of 620 nm using a Sunrise Absorbance Reader (Tecan, Männedorf, Switzerland).

### 4.5. Calculation of the CI

The CI values for the compound combinations were calculated to quantitatively compare the dose–effect relationship of each compound individually and in combination to determine whether a selected combination acts synergistically. CI values were obtained from growth inhibition curves using the constant ratio of the compounds in combination (1:1) as previously described [57]. The CI values were calculated using CalcuSyn software (version 2.0, Biosoft, Cambridge, UK). The Chou Talalay method was adopted to identify antagonism (CI > 1.1), additive effects (CI = 0.9–1.1), or synergism (CI < 0.9) [58].

### 4.6. RT-qPCR

The relative expression of selected genes was determined using RT-qPCR. Total RNA was isolated using a GenElute Mammalian Total RNA Miniprep Kit (Sigma-Aldrich) and reverse transcribed into cDNA as described previously [56]. qPCR was performed in a 10-μL volume using a Kapa Biosystems Quantitative Real-Time PCR kit (Kapa Biosystems, Wilmington, MA, USA) and analyzed using a 7500 Fast Real-Time PCR System and 7500 Software v. 2.0.6 (both obtained from Life Technologies, Carlsbad, CA, USA). To detect differences in the transcript levels among the cell types or after the treatment, Cq values normalized to the endogenous reference control (the GAPDH gene) were compared. The primer sequences used for the ABCB1 and GAPDH genes are provided in Table 4.

### 4.7. Immunoblotting

Protein extracts were obtained using LB1 lysis buffer (50 mM HEPES-KOH, pH 7.5; 140 mM NaCl; 1 mM EDTA; 10% glycerol; 0.5% NP-40; and 0.25% Triton X-100). Total proteins (25 μg/well) were loaded onto 8% (for Pgp analysis) or 10% (for CHEK1 and CHEK2 analysis) polyacrylamide gels and electrophoresed. The separated proteins were then blotted onto PVDF membranes (Bio-Rad Laboratories, Munich, Germany). The membranes were blocked with 5% nonfat dry milk in PBS with 0.1% Tween-20 at RT for 1 h. Subsequently, the blocked membranes were incubated overnight with primary monoclonal antibodies. The following day, the membranes were washed in TBS-Tween and incubated with secondary antibodies at RT for 1 h. The primary and secondary antibodies are listed in Table 5. ECL-Plus detection was performed according to the manufacturer’s instructions (GE Healthcare, Little Chalfont, UK). The obtained protein bands were analyzed using ImageJ software (NIH, MD, USA).

### 4.8. Indirect Immunofluorescence

Indirect fluorescence analysis was performed as described previously [59]. The cells on coverslips were fixed with 3% paraformaldehyde (Sigma-Aldrich) at RT for 20 min. The samples were then permeabilized with 0.2% Triton X-100 (Sigma-Aldrich) in PBS at RT for 1 min. For the DOX and LAMP-2 colocalization analysis, 0.2% Triton was replaced with 100 µM digitonin (Sigma-Aldrich), and the cells were permeabilized at RT for 10 min. The primary and secondary antibodies used for indirect fluorescence staining are listed in Table 6. Coverslips used as negative controls were prepared by omitting the primary antibody. The cell nuclei were counterstained with 0.05% Hoechst 33342 (Life Technologies, Carlsbad, CA, USA). The coverslips with stained cells were mounted using ProLong Diamond Antifade Mountant (Thermo Fisher Scientific, Waltham, MA, USA). For fluorescence evaluation, an Olympus BX-51 microscope was used; the micrographs were captured using an Olympus DP72 CCD camera and analyzed using a Cell^P imaging system (Olympus, Tokyo, Japan).

### 4.9. Cell Cycle Analysis Using Propidium Iodide Staining

The treated cells were harvested from Petri dishes using Accutase (Biosera, Nuaille, France) and fixed in cold 70% ethanol (Sigma-Aldrich) at 4 °C for 30 min. The fixed cells were washed with PBS, and the pellets were carefully resuspended in 100 µL of Vindelov’s staining solution (1 M Tris pH 8 (Sigma-Aldrich); 10 mM NaCl (Sigma-Aldrich); 5 µM ribonuclease A (Sigma-Aldrich); and 75 µM PI) [60]. After 30 min of incubation in Vindelov’s solution at 37 °C, the stained cells were analyzed in a FACSCanto TMII flow cytometer using BD FACS DIVA Software (Beckton Dickinson, CA, USA). The fluorescence of 10,000 cells in each sample was evaluated.

### 4.10. Statistical Analyses

All experiments were performed in biological triplicates (unless otherwise specified). Numerical data are presented as the means ± standard deviation (SD). The data obtained by qPCR or immunoblotting were analyzed using SPSS Statistics software (version 25.0, IBM, New York, NY, USA) by unpaired Welch’s *t*-test followed by the Games–Howell post hoc test. * *p* < 0.05 indicates significant differences compared with the respective control group.

## Figures and Tables

**Figure 1 ijms-23-08549-f001:**
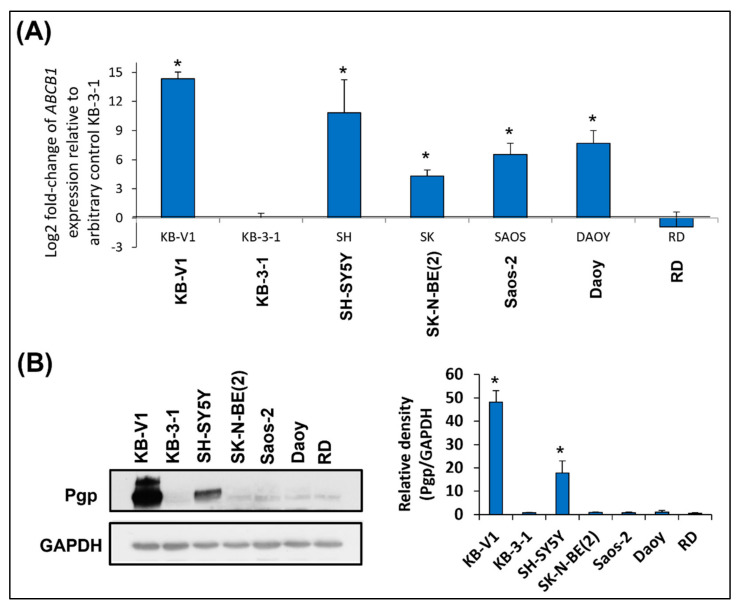
**Measuring Pgp in the selected cell lines.** KB-V1 cells were used as positive controls for *ABCB1* (gene encoding Pgp) expression, whereas KB-3-1 cells were used as negative controls. All experiments were performed in biological triplicates. (**A**) The graph shows the mRNA expression of *ABCB1* (the gene encoding Pgp) in untreated cell lines. The data were obtained using RT-qPCR. The levels of ABCB1 expression are presented as a log2-fold change based on mRNA expression relative to that of the negative control (y = 0). GAPDH served as the reference control. The data were analyzed by one sample *t*-test: * *p* < 0.05 indicates significant differences compared to the respective control group. (**B**) Immunoblot analysis of the endogenous Pgp level in untreated cell lines. GAPDH served as the loading control. The data are presented as the means ± SD and they were analyzed by unpaired *t*-test with Welch’s correction: * *p* < 0.05 indicates significant differences compared to the respective control group.

**Figure 2 ijms-23-08549-f002:**
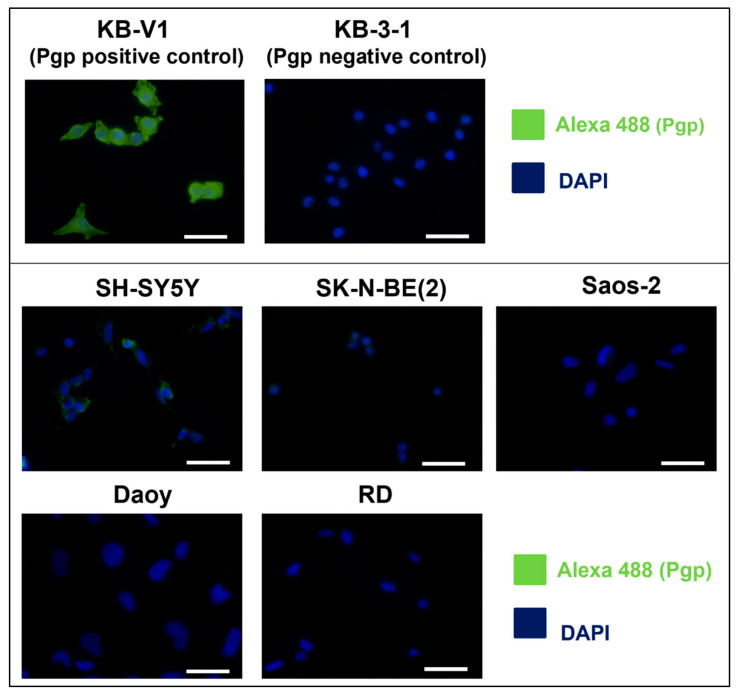
**Pgp expression in the untreated cell lines.** The expression of Pgp analyzed in the untreated cell lines was visualized by indirect immunofluorescence using an anti-Pgp primary antibody and an Alexa-488-conjugated secondary antibody (green signal). Nuclei were labeled by DAPI (blue signal). KB-V1 cells served as positive controls for Pgp expression, whereas KB-3-1 cells served as negative controls. The scale bars represent 50 µm.

**Figure 3 ijms-23-08549-f003:**
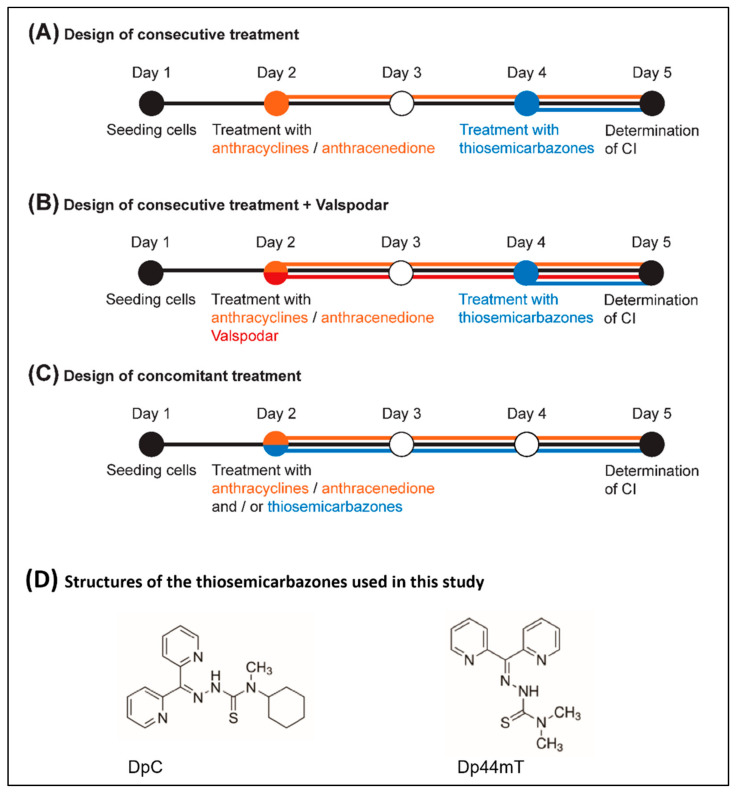
**The three different combination treatment designs used in this study.** (**A**) The design of consecutive treatment included the pretreatment of cells for 48 h with an anthracycline/anthracenedione (either DOX, DAU, or MIT), and then either DpC or Dp44mT was added to the cells undergoing anthracycline/anthracenedione treatment for a period of 24 h. (**B**) To examine whether the mechanism of synergy was based on Pgp transport activity, Valspodar, a selective inhibitor of Pgp, was added to the design of consecutive treatment. (**C**) In the design of concomitant treatment, both anthracycline/anthracenedione (either DOX, DAU, or MIT) and thiosemicarbazones (either DpC or Dp44mT) were added in combination to the cells and proliferation was evaluated after 72 h. (**D**) Structure of di-2-pyridylketone 4-cyclohexyl-4-methyl-3-thiosemicarbazone (DpC) and di-2-pyridylketone 4,4-dimethyl-3-thiosemicarbazone (Dp44mT).

**Figure 4 ijms-23-08549-f004:**
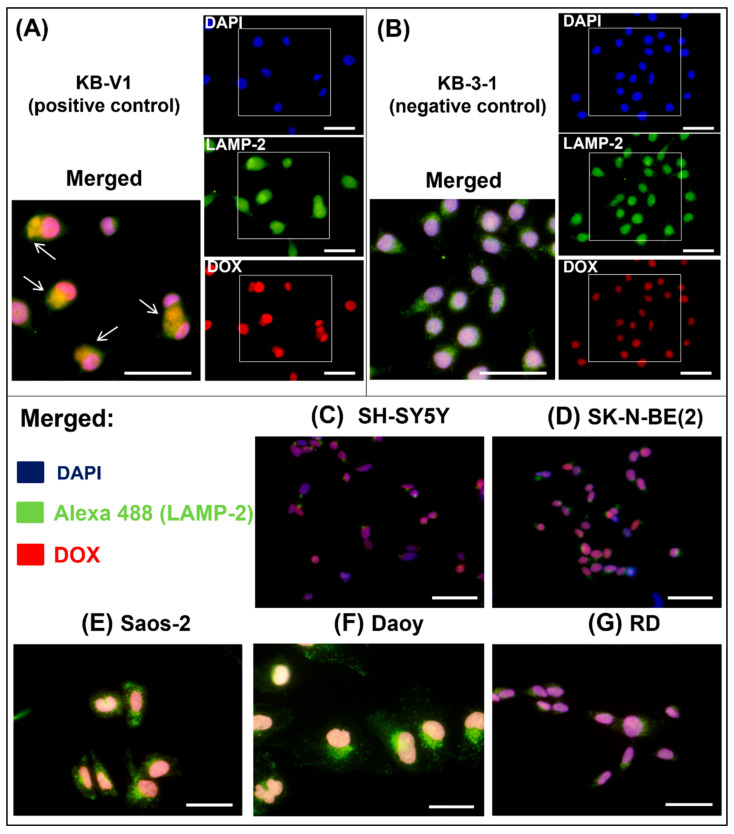
**DOX localization in cells.** All cell lines were treated with 10 µM doxorubicin (DOX). The localization of DOX (red fluorescence) within cells after 2 h of treatment was observed using fluorescence microscopy. Additionally, the target organelles, lysosomes, and nuclei were stained. The lysosomes were visualized by indirect immunofluorescence using an anti-LAMP-2 primary antibody and an Alexa-488-conjugated secondary antibody (green fluorescence). The nuclei were labeled with DAPI (blue fluorescence). The KB-V1 cell line (**A**) served as a positive control for the lysosomal sequestration of DOX (indicated by arrows: yellow fluorescence—overlap of green and red fluorescence emissions). In contrast, the KB-3-1 cells served as a negative control (**B**). The analyses were performed with all five selected cell lines: SH-SY5Y (**C**), SK-N-BE(2) (**D**), Saos-2 (**E**), Daoy (**F**), and RD (**G**) cells. The scale bars represent 50 µm.

**Figure 5 ijms-23-08549-f005:**
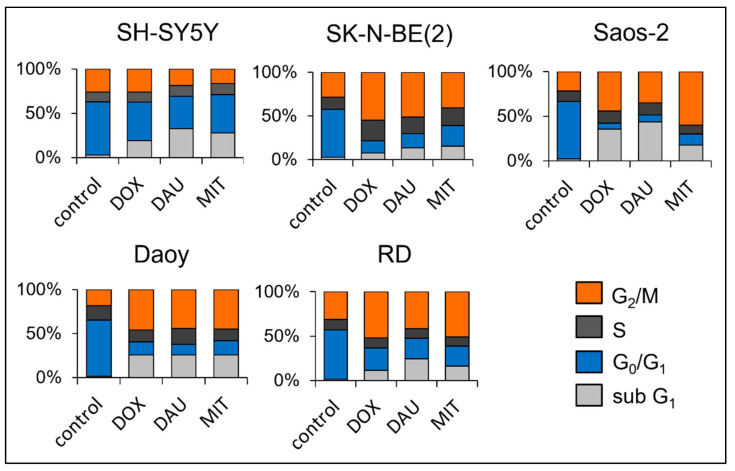
**Analysis of the cell cycle using propidium iodide staining.** Cell cycle analysis of selected cell lines (SH-SY5Y, SK-N-BE(2), Saos-2, Daoy, and RD cells) was performed by quantification of the DNA content with propidium iodide staining and subsequent flow cytometry detection. Changes in the cell cycle were obtained after 72 h of treatment with IC_50_ doses of doxorubicin (DOX), daunorubicin (DAU), or mitoxantrone (MIT). The graphs show the mean proportion of cells (%) in the cell cycle phases: sub G_1_, G_0_/G_1_, S, and G_2_/M. The detailed table showing the means ± SD is provided in Appendix A. All experiments were performed in biological triplicates.

**Figure 6 ijms-23-08549-f006:**
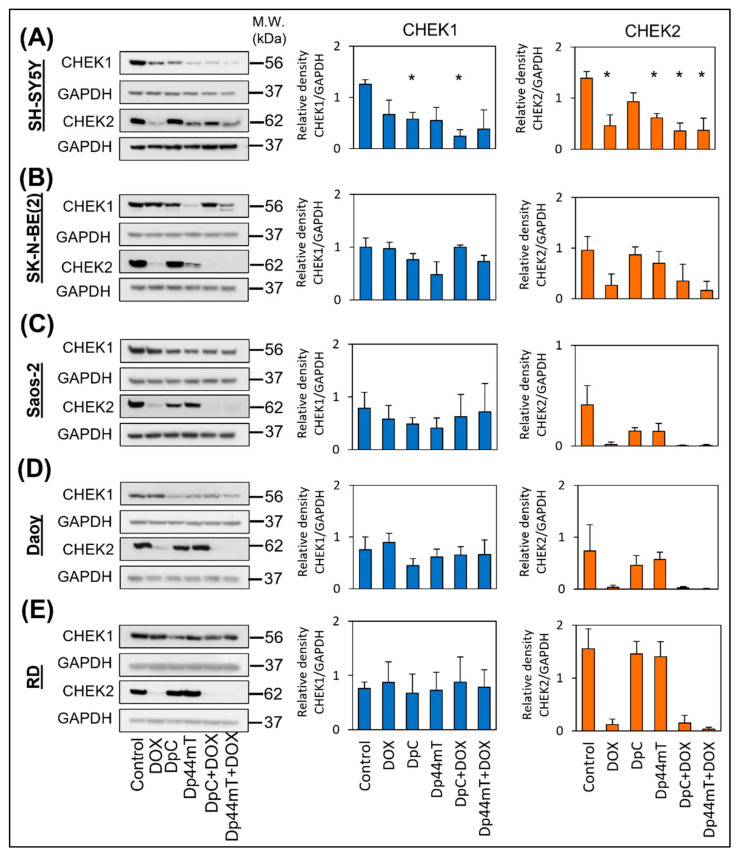
**Immunoblotting to determine CHEK1 and CHEK2 levels in selected cell lines.** CHEK1 and CHEK2 expression was evaluated in the SH-SY5Y (**A**), SK-N-BE(2) (**B**), Saos-2 (**C**), Daoy (**D**), and RD (**E**) cell lines after 3 days of incubation with IC_50_ doses of DOX, DpC, Dp44mT, or their combinations. GAPDH served as the loading control. The experiments were performed in biological triplicates. The obtained data were analyzed by unpaired Welch’s *t*-test followed by Games–Howell post hoc test. * *p* < 0.05 indicates significant differences compared to the respective control group.

**Figure 7 ijms-23-08549-f007:**
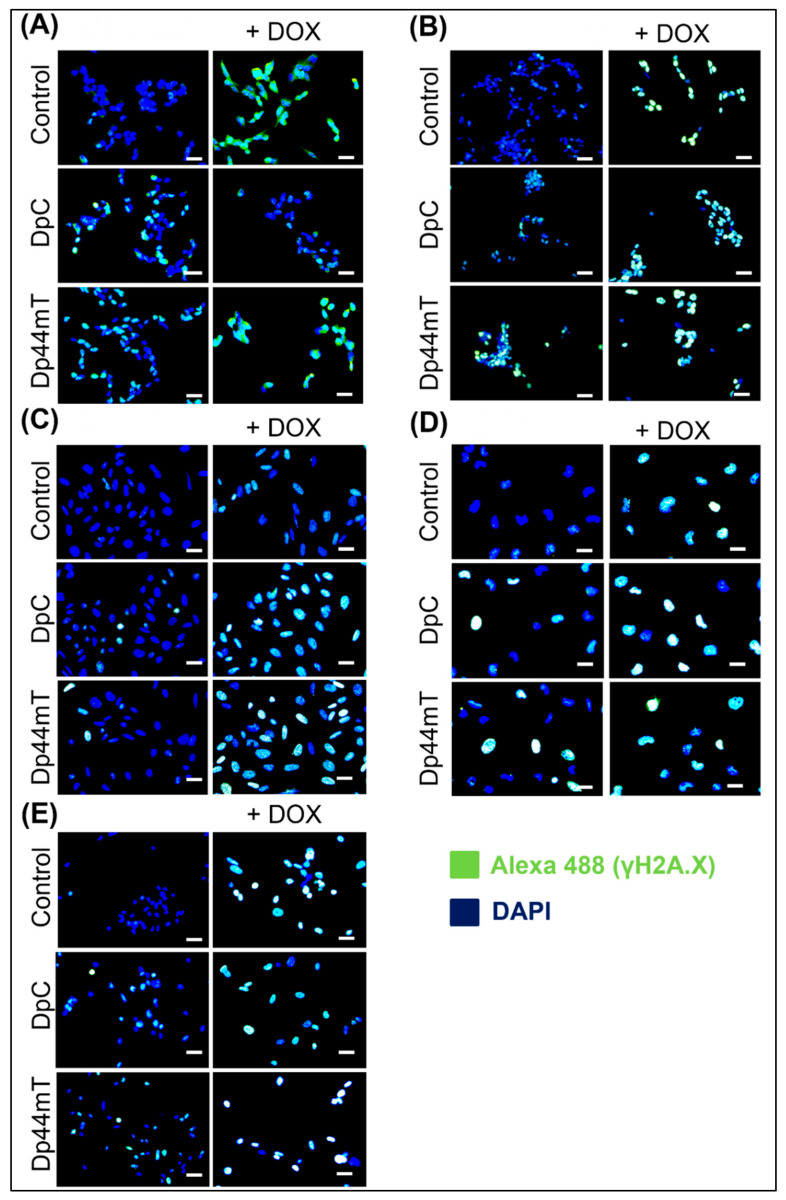
Detection of double-strand breaks (DSBs) in DNA using immunofluorescence staining of phosphorylated γ-H2AX protein. Detection of DSBs in SH-SY5Y (**A**), SK-N-BE(2) (**B**), Saos-2 (**C**), Daoy (**D**), and RD (**E**) cells after 48 h of treatment with IC_50_ doses of DpC, Dp44mT, DOX, or their combination was performed by fluorescence microscopy. Phosphorylated γ-H2AX protein (a DSB marker) was visualized by indirect immunofluorescence using an anti-γ-H2AX primary antibody and Alexa-488-conjugated secondary antibody (green fluorescence). The nuclei were labeled by DAPI (blue fluorescence). The scale bars represent 50 µm.

**Table 1 ijms-23-08549-t001:** IC_50_ values used for subsequent experiments.

Cell Line	DOX [nM] 3D	DAU [nM] 3D	MIT [nM] 3D	DpC [nM] 3D	Dp44m [nM] 3D	DpC [µM] 1D	Dp44mT [µM] 1D
**SH-SY5Y**	29.7 ± 4.9	40.2 ± 13.9	20.1 ± 3.6	3.8 ± 0.3	1.9 ± 0.8	11.1 ± 2.2	30.1 ± 1.5
**SK-N-BE(2)**	40.8 ± 8.1	31.9 ± 8.7	27.0 ± 1.7	7.2 ± 1.2	4.2 ± 1.1	14.8 ± 2.3	31.1 ± 5.2
**Saos-2**	23.9 ± 2.8	20.8 ± 0.9	6.1 ± 0.6	4.4 ± 0.9	6.6 ± 0.9	20.0 ± 4.2	61.6 ± 14.9
**Daoy**	9.8 ± 2.0	17.6 ± 3.3	7.5 ± 0.2	17.0 ± 3.4	7.9 ± 1.9	3.9 ± 0.6	0.9 ± 0.1
**RD**	18.2 ± 3.8	15.7 ± 1.0	26.3 ± 4.4	13.8 ± 2.4	9.4 ± 0.8	11.5 ± 1.5	55.8 ± 7.7

3D indicates cultivation with the respective compound for 3 days (72 h), 1D indicates cultivation for 1 day (24 h).

**Table 2 ijms-23-08549-t002:** Quantitative assessment of the interactions between drugs.

Cell Line	Drug Combination		Treatment Design							
			Consecutive			Consecutive + Valspodar		Concomitant		
			CI		SD	CI	SD	CI	SD	
**SH-SY5Y**	**DOX**	**Dp44mT**	**0.739**		0.064	**0.795**	0.062	**0.779**	0.079	
		**DpC**	**0.793**		0.127	**0.492**	0.068	**0.684**	0.062	
	**DAU**	**Dp44mT**	**0.655**		0.174	**0.828**	0.066	**0.842**	0.124	
		**DpC**	**0.599**		0.077	**0.588**	0.051	**0.647**	0.161	
	**MIT**	**Dp44mT**	**0.507**		0.099	**0.758**	0.121	**0.921**	0.091	
		**DpC**	**0.527**		0.081	**0.898**	0.054	**0.620**	0.222	
**SK-N-BE(2)**	**DOX**	**Dp44mT**	**0.722**		0.053	**0.567**	0.117	**0.712**	0175	
		**DpC**	**0.974**		0.050	**0.635**	0.062	**0.795**	0,161	
	**DAU**	**Dp44mT**	**0.562**		0.118	**0.534**	0.117	**0.732**	0.202	
		**DpC**	**0.640**		0.140	**0.409**	0.074	**0.638**	0.154	
	**MIT**	**Dp44mT**	**0.304**		0.107	**0.524**	0.049	**0.458**	0.157	
		**DpC**	**0.590**		0.137	**0.531**	0.133	**0.569**	0.177	
**Saos-2**	**DOX**	**Dp44mT**	**1.105**		0.102	**0.710**	0.125	**0.657**	0.051	
		**DpC**	**1.205**		0.066	**0.596**	0.057	**0.831**	0.101	
	**DAU**	**Dp44mT**	**1.244**		0.289	**0.712**	0.149	**0.813**	0.135	
		**DpC**	**1.422**		0.085	**0.611**	0.172	**0.672**	0.177	
	**MIT**	**Dp44mT**	**1.108**		0.182	**0.610**	0.056	**0.828**	0.112	
		**DpC**	**1.439**		0.255	**0.723**	0.160	**0.527**	0.108	
**Daoy**	**DOX**	**Dp44mT**	**0.720**		0.210	**0.591**	0.078	**1.055**	0.081	
		**DpC**	**0.578**		0.138	**0.611**	0.015	**0.637**	0.113	
	**DAU**	**Dp44mT**	**0.474**		0.033	**0.602**	0.118	**1.275**	0.280	
		**DpC**	**0.789**		0.167	**0.788**	0.157	**0.854**	0.103	
	**MIT**	**Dp44mT**	**0.663**		0.277	**0.564**	0.137	**0.501**	0.238	
		**DpC**	**0.439**		0.029	**0.746**	0.035	**0.700**	0.175	
**RD**	**DOX**	**Dp44mT**	**0.807**		0.126	**0.472**	0.108	**0.648**	0.043	
		**DpC**	**0.770**		0.008	**0.830**	0.057	**0.712**	0.312	
	**DAU**	**Dp44mT**	**0.458**		0.143	**0.383**	0.059	**0.773**	0.211	
		**DpC**	**0.609**		0.097	**0.671**	0.173	**0.764**	0.118	
	**MIT**	**Dp44mT**	**0.501**		0.099	**0.366**	0.031	**0.493**	0.108	
		**DpC**	**0.641**		0.081	**0.541**	0.134	**0.638**	0.127	
**Categories of Interactions**										
0.31–0.70	synergism				0.91–1.10		nearly additive			
0.71–0.85	moderate synergism				1.11–1.20		slight antagonism			
0.86–0.90	slight synergism				1.21–1.45		moderate antagonism			

Computational analysis of the interaction of doxorubicin (DOX), daunorubicin (DAU), or mitoxantrone (MIT) with thiosemicarbazones (DpC or Dp44mT) was performed using Calcusyn software. The combination index (CI) was calculated from the growth inhibition curves of the compounds alone or their combinations. A ratio of 1:1 was maintained between the drugs added in combination. The method of Chou and Talalay was employed to define synergism, additive effects, or antagonism. The experiments were performed in biological triplicates.

**Table 3 ijms-23-08549-t003:** Composition of culture media used for culturing the selected cell lines.

Cell Line	Type of Medium	Glucose	FCS	Atb	Glu	NEAA
Daoy	DMEM	Low	10%	+	+	+
KB-3-1	DMEM	High	10%	+	+	-
KB-V1	DMEM + VBL	High	10%	+	+	+
RD	DMEM	High	10%	+	+	+
Saos-2	DMEM	Low	10%	+	+	-
SH-SY5Y	DMEM:F12 (1:1)	High	20%	+	+	+
SK-N-BE(2)	DMEM:F12 (1:1)	High	20%	+	+	+

DMEM—Dulbecco’s modified Eagle’s medium; low glucose—1000 mg/L; high glucose—4500 mg/L; HVBL—vinblastine 1 µg/mL; FCS—fetal calf serum; Atb—streptomycin (100 μg/mL) and penicillin (100 IU/mL); Glu—2 mM L-glutamine; NEAA—1% nonessential amino acids.

**Table 4 ijms-23-08549-t004:** Sequences of the primers used for RT-qPCR.

Gene	Primer Sequence	Product Length (bp)
** *ABCB1* **	F: 5′-CTTTAGTGGAAAGACCACAGATGA-3′ R: 5′-CTTTAGTGGAAAGACCACAGATGA-3′	228
** *GAPDH* **	F: 5′-AGC CAC ATC GCT CAG ACA CC-3′ R: 5′-GTA CTC AGC GCC AGC ATC G-3′	302

F—forward; R—reverse.

**Table 5 ijms-23-08549-t005:** Primary and secondary antibodies used for immunoblotting.

Primary antibodies					
Antigen	Type/Host	Clone	Catalog No.	Manufacturer	Dilution
GAPDH	Mono/Rb	14C10	2118S	CST	1:10,000
CHEK1	Mono/Mo	2G1D5	2360S	CST	1:1000
CHEK2	Mono/Rb	D9C6	6334S	CST	1:1000
Pgp	Mono/Mo	F4	P7965	Sigma-Aldrich	1:5000
**Secondary antibodies**					
**Host**	**Specificity**	**Conjugate**	**Catalog No.**	**Manufacturer**	**Dilution**
Goat	Anti-Rb IgG	HRP	7074	CST	1:5000
Horse	Anti-Mo IgG	HRP	7076	CST	1:5000

Mono—monoclonal; Rb—rabbit; Mo—mouse; CST—Cell Signaling Technology Inc.; Bioss—Bioss Antibodies, Inc.

**Table 6 ijms-23-08549-t006:** Primary and secondary antibodies used for indirect immunofluorescence staining.

Primary antibodies					
Antigen	Type/Host	Clone	Catalog No.	Manufacturer	Dilution
Pgp	Mono/Mo	F4	P7965	Sigma-Aldrich	1:100
LAMP-2	Mono/Mo	H4B4	ab25631	Abcam	1:100
Phospho-γH2A.X	Mono/Mo	3F2	MA1-2022	Invitrogen	1:100
**Secondary antibodies**					
**Host**	**Specificity**	**Conjugate**	**Catalog No.**	**Manufacturer**	**Dilution**
Donkey	Anti-Mo IgG	AF-488	A21202	Invitrogen	1:200

Mono—monoclonal; Mo—mouse.

## Data Availability

The data presented in this study are available in the article and the Appendix A.

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
