# Peer review of "Thiosemicarbazones Can Act Synergistically with Anthracyclines to Downregulate CHEK1 Expression and Induce DNA Damage in Cell Lines Derived from Pediatric Solid Tumors"

_ijms, 2022, doi:10.3390/ijms23158549_

Round 1
Reviewer 1 Report
Veselska and coworkers present the thiosemicarbazone synergy with anthracycline in pediatric solid tumors. The authors have demonstrated the efficacy of two thiosemicarbazones in combination with anthracyclines against cancer cells derived from pediatric solid tumors. In addition, new insight into the mechanism of synergistic interaction between anthracyclines and thiosemicarbazones is provided. The authors suggest DNA damage and the inhibition of checkpoint kinases as another mechanism of synergy. Given the challenges in the treatment of MDR in pediatric cancers, the mechanistic information presented here is interesting. The experiments were carefully carried out and the results are presented appropriately. The manuscript merits publication in IJMS after addressing the below comments.
1. Please include the structures of DpC and Dp44mT.
2. In the introduction- the authors discuss the side effects of anthracyclines. However, there is no experimental evidence included in the results section that supports if the combination of thiosemicarbazones with anthracyclines addresses the cytotoxicity issue.
3. Section 2.1- The authors have provided the origin of KB-V1 cells and KB-3-1 (i.e., cervical carcinoma). Similarly, please provide the details origin of the other cell lines.
4. The title of the manuscript states “Novel thiosemicarbazones can act synergistically with anthracyclines to downregulate CHEK1 expression and induce DNA damage in cell lines derived from pediatric solid tumors” However, the compounds used in this study are not new. They have been previously reported. The title needs to be modified. Furthermore, the authors should make this change wherever is applicable. 5. Figure 6- combined treatment of thiosemicarbazones with DOX did not show a decrease in CHEK1 levels in SK-N-BE(2), Saos-2, and Daoy cell lines. Can the authors comment on this observation?
Reviewer 2 Report
The manuscript “Novel thiosemicarbazones can act synergistically with anthracyclines to downregulate CHEK1 expression and induce DNA damage in cell lines derived from pediatric solid tumors” Paukovcekova et al. investigated the synergistic interaction between anthracyclines and novel thiosemicarbazones in model cell lines derived from pediatric solid tumors.
They concluded that overcoming drug resistance mediated by Pgp is not the main mechanism of the observed synergism. In fact, another synergistic mechanism based on enhanced DNA damage and the inhibition of checkpoint kinases was suggested.
Since the acquisition of multidrug resistance (MDR) is currently the leading cause of treatment failure in pediatric oncology, the topic is interesting and up-to-date.
The manuscript is written clearly and the presentation of results follows a coherent line; various methods and techniques are used. However, I have only few comments on the current version of the manuscript to share with the Authors.
Results section:
The Authors should add p values in Figure 1
Discussion section:
The Authors investigated the expression profile of Pgp in different cancer cell lines. Among these cell lines, is there a difference in the expression of the tumor suppressor p53? Is the up-regulation of Pgp dependent or independent of p53? I suggest to argue this aspect; it could be beneficial for the discussion.
Quality of English language:
The manuscript should be English proofread. Some typo are present.
